# Laser Therapy for Cutaneous Kaposi Sarcoma: A Systematic Review and Meta-Analysis

**DOI:** 10.3390/cancers17223708

**Published:** 2025-11-20

**Authors:** Tomer Mimouni, Meital Oren-Shabtai, Aviv Barzilai, Sharon Baum, Yehonatan Noyman, Shohat Michael, Riad Kassem

**Affiliations:** 1Gray Faculty of Medical and Health Sciences, Tel Aviv University, Tel Aviv 6997801, Israel; mimouni1@mail.tau.ac.il (T.M.);; 2Division of Dermatology, Rabin Medical Center, Beilinson Hospital, Petach Tikva 4941492, Israel; 3Dermatology Department, Sheba Medical Center, Ramat-Gan 5262000, Israel; 4Clalit Esthetic Clinic, Clalit Health Services, Petach Tikva 4941492, Israel

**Keywords:** Kaposi sarcoma, cutaneous neoplasms, laser therapy, Nd:YAG, pulsed dye laser, indocyanine green laser

## Abstract

Cutaneous Kaposi sarcoma is a vascular malignancy characterized by abnormal blood vessel growth in the skin. Although many local treatment options are available, none are considered the standard of care, and some may cause long-term side effects. Laser therapy is a modern modality that selectively targets abnormal vessels while sparing most of the surrounding tissues. In this study, we systematically reviewed all available data on the use of laser treatment for cutaneous Kaposi sarcoma lesions. Laser therapy, particularly Neodymium:yttrium–aluminum–garnet laser (Nd:YAG), is generally effective in treating lesions and is well tolerated, with only mild and temporary side effects. However, recurrence after treatment is rare. These results suggest that laser therapy could provide patients with a safer option for controlling their disease, while also achieving desirable cosmetic results.

## 1. Introduction

Kaposi sarcoma (KS) is a vascular malignancy associated with human herpesvirus 8 (HHV-8) that may present with lesions involving the skin, mucous membranes, lymph nodes and other visceral organs [1,2].Cutaneous lesions of KS often appear as violaceous macules that progress to plaques and nodules, potentially leading to pain, bleeding, and cosmetic concerns [3,4]. There are four epidemiologic subtypes of KS: classic KS, which predominantly occurs in older Mediterranean or Eastern European males; endemic KS, a more aggressive subtype affecting people living in sub-Saharan Africa; iatrogenic KS, described mainly in patients receiving immunosuppressive therapy; and epidemic KS, also known as HIV-associated KS. Although there are clinical and demographic differences between these subtypes of KS, cutaneous involvement remains a clinical feature shared by all and represents the most significant aspect of morbidity associated with KS [2].

There is no consensus regarding the standard of care for cutaneous KS. Local treatment includes surgical excision, cryotherapy, radiotherapy, intralesional chemotherapy, and laser therapy, aiming to relieve any symptomatic features of cutaneous KS, and achieve local control of lesions. Systemic therapy, including chemotherapy, interferons, and antiangiogenic agents, is mostly considered for patients with widely disseminated or aggressive disease, whereas local treatments are used for patients with more localized involvement [5].

Laser therapy is considered a targeted local treatment modality for KS because it may selectively target vascular lesions that minimally damage the surrounding skin [6,7]. Reported laser modalities include carbon dioxide (CO_2_), argon, pulsed dye laser (PDL), Q-switched alexandrite laser, Q-switched ruby laser, pro yellow laser (PYL), neodymium-doped yttrium-aluminum-garnet (Nd:YAG), and indocyanine green (ICG) [7,8,9,10,11,12,13,14]. Few studies have been published for each laser modality, mostly reporting partial and heterogeneous protocols. This variability limits comparability across studies and prevents definitive conclusions regarding the efficacy of laser therapy for cutaneous KS.

To address this gap, we conducted a systematic review and meta-analysis of the efficacy and safety of laser modalities for cutaneous KS, to clarify the potential role of laser therapy as a local treatment option.

## 2. Materials and Methods

### 2.1. Protocol and Reporting

This systematic review and meta-analysis was conducted in accordance with the Preferred Reporting Items for Systematic Reviews and Meta-Analyses (PRISMA) guidelines [15], and the completed PRISMA checklist is provided in Appendix A. The protocol for this review was developed a priori and registered in the PROSPERO database (registration number: CRD420251125463).

### 2.2. Search Strategy

A comprehensive literature search with no date limits was performed in August 2025 using multiple electronic databases including MEDLINE (PubMed), EMBASE, Cochrane Library, ClinicalTrials.gov, and NIH RePORTER. The search strategies included Medical Subject Headings (MeSH) and free-text terms for “Kaposi Sarcoma” and “Laser”, with modifications made to the strategies for each database. Reference lists from eligible articles and previous reviews were manually screened to identify potentially eligible studies.

### 2.3. Eligibility Criteria

Studies that met the following criteria were included: (1) Design—randomized controlled trials (RCTs), prospective or retrospective cohort studies, or case series consisting of ≥3 patients; (2) Population—patients with a clinical and/or histopathological diagnosis of solely cutaneous KS; (3) Intervention—laser therapy of any wavelength, pulse duration, or treatment protocol; (4) Comparison—no intervention, placebo or other local treatments for cutaneous KS; (5) Outcomes—complete or partial remission of KS lesions, recurrence rates of treated lesions, and adverse effects of treatment; (6) Language—no language restrictions; non-English studies were translated when possible.

The exclusion criteria were as follows: (1) case reports or case series of <3 patients; (2) studies that did not exclusively include cutaneous localized KS; (3) studies examining combination therapies not enabling the isolation of the effect of the laser device; (4) studies that did not provide relevant clinical outcomes; and (5) duplicate publications or overlapping cohorts, including cases of multiple publications from the same cohort, of which the latest or most complete dataset would be included as per the protocol.

### 2.4. Outcomes

The primary outcome was complete response (CR), defined as total or near-total disappearance of the treated lesions. Secondary outcomes included (1) partial response (PR), defined as a documented clinical reduction of >50% in lesion size; (2) recurrence rate, defined as regrowth of previously treated lesions; and (3) treatment-related adverse events, assessed by frequency, type, and severity. Additional variables extracted included study design, patient demographics, lesion characteristics, laser parameters, and follow-up duration.

### 2.5. Study Selection and Data Extraction

Two researchers (T. Mimouni and M. Oren-Shabtai) independently screened all titles and abstracts. The full texts of the potentially eligible studies were evaluated according to the inclusion criteria. Publications were graded according to their level of evidence: (A) randomized controlled trial (RCT), (B) prospective study, and (C) retrospective study. Disagreements were resolved by consensus or by a third reviewer (R. Kassem). Mimouni extracted the data in electronic form and Oren-Shabtai reviewed the data.

The risk of bias in the included studies was independently rated by two reviewers using the Cochrane RoB 2.0 tool for randomized trials and the Newcastle–Ottawa Scale for observational studies [16,17]. Disagreements were resolved through discussion or a third reviewer (R. Kassem).

### 2.6. Data Analysis and Synthesis

For pooled analysis, treatment responses were standardized according to Junejo (2024). A significant response (SR) was defined as substantial (>50%) reduction in lesion color intensity, size or texture of the lesion [18]. Studies that descriptively reported outcomes without quantifiable response rates were excluded from the meta-analysis. SR rates and 95% confidence intervals (CIs) were pooled and calculated using the inverse variance method following logit transformation. We performed analyses using the DerSimonian and Laird random effects, with Hartung–Knapp adjustment for the CIs, given the few studies included. Sensitivity analysis was performed using a leave-one-out approach. Heterogeneity was assessed visually based on the forest plot examining non-overlapping CIs and using the chi-square (χ^2^) statistic, with *p* < 0.05 indicating significant heterogeneity and *I*^2^ > 50% indicating considerable heterogeneity. Statistical analyses were conducted using R software (version 4.5.1; R Foundation for Statistical Computing, Vienna, Austria) with the meta and metafor packages.

## 3. Results

### 3.1. Study Selection

A total of 351 records were identified through the database search. After removing duplicates and screening titles and abstracts, 27 full-text articles were assessed for eligibility. Of these, 19 studies were excluded for the following reasons: lack of relevant data (*n* = 10), fewer than three patients (*n* = 8), and interventions not limited to laser therapy (*n* = 1). Finally, eight studies met the inclusion criteria and were included in this systematic review. The study selection process is summarized in the PRISMA flow diagram (Figure 1).

### 3.2. Study Characteristics

The included studies were published between 1992 and 2024 and conducted in Germany, Italy, Turkey, and the United States. Sample sizes ranged from 3 to 30 patients, yielding a total of 79 patients and 371 treated lesions, of which three patients were eventually lost to follow-up. One study was an RCT (grade A), four were prospective (grade B) and three were retrospective (grade C). The median age of patients at the time of treatment ranged from 33 to 83 years. Males comprised the majority of participants, accounting for 88.6% of the studies. The most common Kaposi sarcoma (KS) subtypes were HIV-associated KS (*n* = 41, 51.9%) and classic KS (*n* = 37, 46.8%), with one iatrogenic case reported (Junejo, 2024) [18]. Baseline lesion sizes varied widely, from sub-centimeter papules to nodules measuring up to 15 cm in diameter. A summary of the study characteristics is provided in Table 1 [7,10,14,18,19,20,21,22].

### 3.3. Risk of Bias

Seven observational studies were evaluated using the Newcastle–Ottawa Scale [17]. Excluding Silvestri (2022) [21], which was assessed as high-quality, the remaining six observational studies were designated as moderate-quality. The single RCT by Tappero (1992) [10] was evaluated using Cochrane RoB 2.0 [16] and was found to be at moderate risk with some concerns. The results are summarized in Table 1.

### 3.4. Laser Modality and Treatment Parameters

Several types of lasers were used in the included studies. The most frequently reported laser was neodymium:yttrium–aluminum–garnet (Nd:YAG, 1064 nm), followed by indocyanine green (ICG, 805 nm) and pulsed dye laser (PDL, 585–595 nm). Junejo (2024) described the alternating use of PDL and Nd:YAG in different sessions in a single patient. Treatment parameters varied considerably, with fluence ranging from 90 to 260 J/cm^2^ for Nd:YAG and ICG, and 7–9.5 J/cm^2^ for PDL, and pulse durations spanning 0.45–25 ms. The interval between sessions was typically 4–6 weeks, with the total number of sessions ranging from 1 to 12. The follow-up duration varied substantially, from 2.8 months to >61 months. A summary of laser modalities and parameters is provided in Table 2 [7,10,14,18,19,20,21,22].

### 3.5. Treatment Outcomes

The efficacy outcomes varied substantially across the included studies (Table 2), showing heterogeneity in both outcome definitions and reporting methods. Among the studies using ICG, one reported 31.6% CR and 68.4% PR, whereas another demonstrated 93.3% CR with no PR. A single study exclusively used PDL and documented an overall response rate of 44% (combined CR + PR). In a study by Junejo (2024), outcomes differed by treatment modality: PDL monotherapy achieved 0% CR with a 54.1% SR, Nd:YAG monotherapy achieved a 100% significant response, and combined PDL plus Nd:YAG therapy resulted in 100% CR. Two studies that employed Nd:YAG alone reported differing outcomes, with one achieving 100% CR and the other achieving 26.7% CR and 53.3% PR. Finally, two additional studies described clinical and dermoscopic improvements qualitatively without providing CR or PR rates [7,10,14,18,19,20,21,22].

A notable byproduct of the treatment was a reduction in lymphedema. Although it is not a predefined outcome in most trials, this improvement has been described by Bostanci (2024), Nasca (2020), and Özdemir (2017) [7,19,20].

Most studies noted no recurrence during follow-up, with Tappero (1992) being the only study that reported a 100% recurrence rate through a 2.8-month follow-up period using PDL. Six patients showed signs of disease progression and required systemic treatment. Importantly, no progression from cutaneous to mucosal manifestations was observed, and no deaths occurred during the follow-up period. Adverse events attributable to therapy were generally mild with reports of minimal atrophic scarring, temporary post-inflammatory hyperpigmentation, blistering, and transient edema; none of the events were severe enough for treatment discontinuation. Notably, blistering and transient edema have been observed only with PDL, whereas other adverse effects were reported across all laser modalities [7,10,14,18,19,20,21,22].

### 3.6. Meta-Analysis

We excluded the studies by Bostanci (2024) and Özdemir (2017) from the meta-analysis because of the descriptive form of their outcome reports (see Figure 1 and Table 1) [7,19]. The pooled analysis of SR rates, defined as >50% improvement in lesion parameters, was 88% (95% CI: 44.1–98.6%, Figure 2), although there was a considerable amount of heterogeneity in the studies (*I*^2^ = 86.3%). A leave-one-out sensitivity analysis showed that exclusion of any single study did not substantially alter the pooled remission rate (range: 81.2–92.3%), indicating that the results were robust.

## 4. Discussion

This systematic review and meta-analysis evaluated the efficacy and safety of laser-based therapies for cutaneous KS and presents the first unified quantitative synthesis to provide a pooled estimate of clinical response to laser therapy for cutaneous KS. Eight studies published over the past 33 years encompassing three laser types (Nd:YAG, ICG and PDL) were included. In total, 76 patients with 371 treated lesions completed the follow-up period and were analyzed [7,10,14,18,19,20,21,22].

The included studies demonstrated considerable variability in outcome reporting. Ther CR rates ranged from 5.9% to 100%, whereas PR was reported in several studies (45.1–68.4%). One study reported the combined CR + PR rate, and two studies described clinical and dermoscopic improvements in treated lesions. When outcomes were standardized to a uniform definition, the pooled SR rate was high at 88% (95% CI: 44.1–98.6%, *I*^2^ = 86.3%).

Differences in the reported outcomes may be explained by the diversity of laser modalities employed and variations in treatment protocols, including fluence, pulse duration, and pulse delay. Notably, PDL and ICG were more commonly used in the 1990s and the early 2000s [10,14,22], whereas recent studies have predominantly employed Nd:YAG [7,18,19,20,21]. A major finding of this systematic review was the absence of recurrence among all lesions treated with Nd:YAG. Notably, despite its higher efficacy and greater penetration depth than PDL, Nd:YAG was not associated with a higher incidence of adverse events. Additionally, an advantageous byproduct of the use of Nd:YAG achieved in three patients across three different studies was a reduction in lymphedema [7,19,20]. PDL operates at wavelengths between 585 and 600 nm, limiting its penetration depth to approximately 1.2 mm [23] and restricting its efficacy in treating nodular or deeper KS lesions. In contrast, Nd:YAG operates at a wavelength of 1064 nm, increasing its penetration depth to 5 mm [24]. Furthermore, Nasca (2020) reported the delivery of ND:YAG through a tilted angle to optimize the targeting of deeper vessels in the basal layers of lesions, further minimizing damage to the surrounding tissues and achieving a 100% CR rate with no recurrence [20]. The use of ICG in the treatment of cutaneous KS remains relatively limited and experimental [14,22], thus no definitive conclusions can be drawn regarding its effectiveness or safety.

Additional discrepancies likely reflect heterogeneity in the characteristics of the treated lesions, as most studies did not stratify the outcomes by lesion type or size. Another contributing factor may be the higher vascular density typically observed in HIV-associated KS than in classic KS, which could result in greater laser absorption and enhanced therapeutic effects [25]. This finding may support the treatment of patients with HIV-associated KS with Nd:YAG, where deeper penetration and vascular selectivity may maximize therapeutic efficacy.

Radiotherapy, as a local therapy for cutaneous KS, has reported CR rates ranging from 32% to 99% [26,27,28,29,30]. Nevertheless, its adverse effect profile is far less favorable; there are reports of patients developing acute reactions of erythema, dermatitis, and even ulceration as well as an increasing possibility of chronic sequelae, such as sclerosis, morphea, permanent atrophy and hair loss [28,30,31,32]. Cryotherapy has also been described as a palliative option, particularly for small (<1 cm), superficial, or patchy lesions [32,33]. In contrast, laser therapy, particularly Nd:YAG, has been shown to have high efficacy and a preferable safety profile with adverse effects that are typically mild, temporary, and localized only to minimal scarring or pigmentation change. Severe localized tissue destruction or long-term sequelae caused by laser therapy have not been reported. These findings support laser therapy as a safe modality for select patients.

Our meta-analysis resulted in substantial heterogeneity (*I*^2^ = 86.3%), and the accompanying 95% CI was wide (44.1–98.6%) due to the small sample sizes, low number of studies included, and varied outcomes among the studies. Additionally, the present study was limited by the variance in laser modalities, outcome definitions, moderate quality of the evaluated publications and heterogeneity across studies in methods and outcomes. Excluding one RCT, all the studies included in this systematic review were observational, thus limiting the level of evidence provided.

Before laser therapy can be recommended as the standard of care for the treatment of cutaneous KS, further research is necessary and should focus on large, prospective and standardized trials directly comparing laser modalities versus other local treatments. Additionally, exploring optimal fluence, pulse duration and treatment intervals could improve the understanding and refinement of clinical protocols.

## 5. Conclusions

In conclusion, laser therapy represents a promising, targeted and minimally invasive option for managing cutaneous KS with a high pooled response rate, minimal recurrence, and a favorable safety profile. Among the available modalities, Nd:YAG laser demonstrated the most consistent balance of efficacy, depth of penetration, and safety profile. While current evidence supports its use for localized lesions with good cosmetic outcomes, standardized treatment protocols and larger, controlled studies are still required before laser therapy can be formally integrated into clinical guidelines.

## Figures and Tables

**Figure 1 cancers-17-03708-f001:**
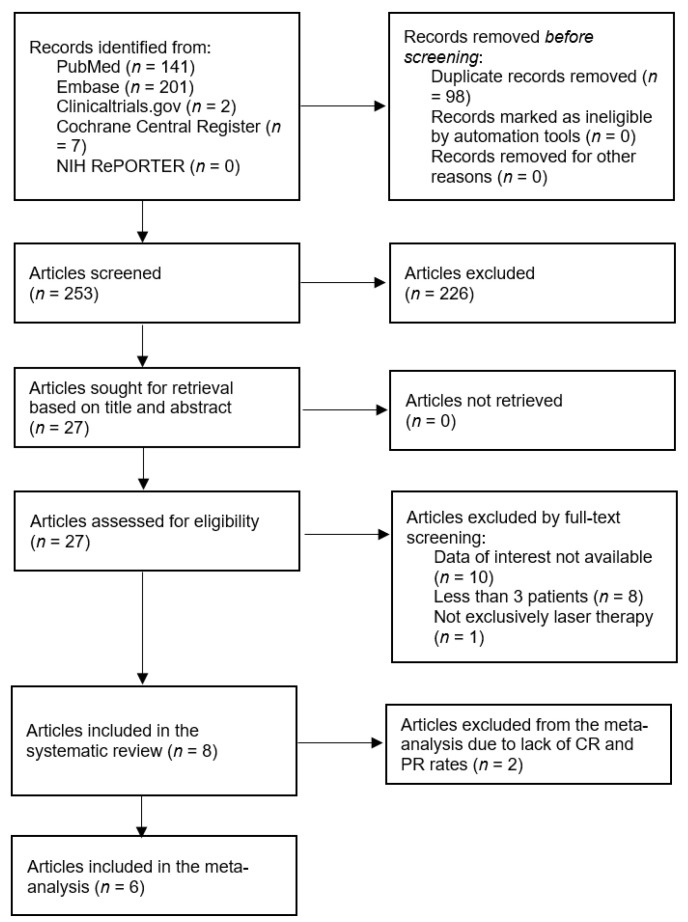
A systematic literature search was performed according to PRISMA guidelines. Preferred Reporting Items for Systematic Reviews and Meta Analyses.

**Figure 2 cancers-17-03708-f002:**
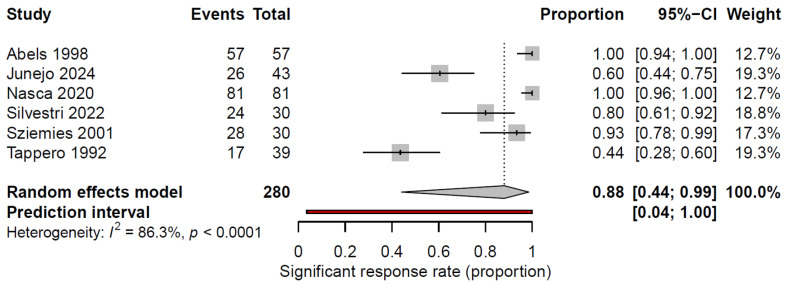
Forest plot summarizing the significant response rate (>50% reduction in color intensity, size or texture of lesions) to laser therapy for cutaneous KS lesions. Significant response rates and 95% CI are presented for each study. The pooled significant response rate is represented by the diamond at the bottom [10,14,18,20,21,22].

**Table 1 cancers-17-03708-t001:** Baseline characteristics and quality assessment of the included studies.

Author, Publication Year	Country	Study Design	Age at Treatment, Years, Median (Range)	Male, *n* (%)	HIV Positive, *n* (%)	KS Subtype, (*n*, %)	Sample Size, *n* (Lesion *n*)	Lesion Type	Lesion Sizes at Treatment, Diameter Range/Area Range	Risk of Bias ***
Abels, 1998 [14]	Germany	Observational	33 (31–67)	3 (100)	3 (100)	HIV-associated (3, 100)	3 (57)	Macules, plaques, nodules	0.4–2 cm	Moderate
Bostanci, 2024 [19]	Turkey	Observational	76 (57–78)	2 (66.7)	0 (0)	Classic (3, 100)	3 (42)	Nodules	0.3–12 cm	Moderate
Junejo, 2024 [18]	USA	Observational	42 (35–91)	5 (83.3)	3 (50)	Classic (3, 50), HIV-associated (2, 33.3), Iatrogenic (1, 16.7)	5 * (43)	Papules, plaques	NA	Moderate
Nasca, 2020 [20]	Italy	Observational	83 (64–86)	7 (77.8)	NA	Classic (9, 100)	9 (81)	Nodules	0.5–3 cm	Moderate
Özdemir, 2017 [7]	Turkey	Observational	65 (45–83)	4 (57.1)	0 (0)	Classic (7, 100)	7 (49)	Papulonodular	1–15 cm	Moderate
Silvestri, 2022 [21]	Italy	Observational	57.5 (34–76)	28 (93.3)	15 (50)	Classic (15, 50), HIV-associated (15, 50)	30 (30)	Macules, papules, plaques, nodules	0.5–11.1 cm	Low
Szeimies, 2001 [22]	Germany	Observational	52.5 (33–65)	6 (100)	6 (100)	HIV-associated (6, 100)	6 (30)	Macules, plaques, nodules	0.16–6 cm^2^	Moderate
Tappero, 1992 [10]	USA	RCT	36 (26–45)	15 (100)	15 (100)	HIV-associated (15, 100)	13 ** (39)	Papules	21–247 cm^2^	Some concerns (moderate)

* Junejo was originally *n* = 6, due to loss to follow-up *n* = 5. ** Tappero was originally *n* = 15, due to loss to follow-up *n* = 13. *** Using the Cochrane RoB 2.0 scale for RCTs and the Newcastle–Ottawa Scale for observational studies.

**Table 2 cancers-17-03708-t002:** Laser therapy characteristics and efficacy outcomes of the included studies.

Author, Publication Year	Laser Type	Wavelength, nm	Laser Size Range, cm	Fluence Range, J/cm^2^	Pulse Protocol	Number of Treatment Sessions	Treatment Sessions Intervals, Weeks	CR Rate (%)	PR Rate (%)	Recurrence Rate (%)	Follow-Up Duration, Months
Abels, 1998 [14]	ICG	805	2	100	0.5–6 W/cm^2^ pulse power	1	NA	18/57 (31.6)	39/57 (68.4)	0/57 (0)	0/57 6–15
Bostanci, 2024 [19]	Nd:YAG	1064	0.3–0.7	200–250	10–20 ms pulse duration	1–2	4	NA, clinical and dermoscopic improvements of all lesions	NA	0/3 (0)	12
Junejo, 2024 [18]	PDL (N = 3), Nd:YAG (N = 1), PDL + Nd:YAG (N = 1)	1064 (Nd:YAG), 595 (PDL)	0.5 (Nd:YAG), 0.7–1 (PDL)	90–140 (Nd:YAG), 7–9.5 (PDL)	20–25 ms pulse duration (Nd:YAG), 1.5–40 pulse duration (PDL)	2–12	6	* PDL: 0/37 (0), Nd:YAG: 0/3 (0), PDL + Nd:YAG: 100 (3/3)	* PDL: 20/37 (54.1), Nd:YAG: 3/3 (100), PDL + Nd:YAG: 0 (0/3)	0/43 (0)	7–61
Nasca, 2020 [20]	Nd:YAG	1064	0.5–0.7	14–200	5 ms pulse duration, triple pulse, 10 ms delay	1–2	4	81/81 (100)	0/81 (0)	0/81 (0)	12
Özdemir, 2017 [7]	Nd:YAG	1064	0.4–0.6	180–260	1.5 ms pulse duration, 5 ms delay	1–4	4	NA, clinical and dermoscopic improvement of all lesions	NA	0/49 (0)	6
Silvestri, 2022 [21]	Nd:YAG	1064	0.25–0.5	120–140	3 ms pulse duration, double pulse, 20 ms delay	4	4	8/30 (26,7)	16/30 (53.3)	0/24 (0)	2.8
Szeimies, 2001 [22]	ICG	805	2	100	3 W/cm^2^ pulse power, 33 s per lesion	1	NA	28/30 (93.3)	NA	0/28 (0)	24
Tappero, 1992 [10]	PDL	585	0.5	8–9.25	0.45 ms pulse duration	2–4	4–6	17/39 ** (44)		17/17 (100)	2.8

* Junejo categorized responses to treatment as complete response (76–100% improvement) and significant response (51–75% improvement), with *n* = 3 and *n* = 23, respectively. ** Tappero reported CR and PR rates combined.

## Data Availability

No new data were created or analyzed in this study.

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
