# Peer review of "Laser Therapy for Cutaneous Kaposi Sarcoma: A Systematic Review and Meta-Analysis"

_cancers, 2025, doi:10.3390/cancers17223708_

Round 1
Reviewer 1 Report
Comments and Suggestions for Authors
Dear authors
-
Limited evidence
-
The quantitative synthesis includes only six studies (79 patients, 371 lesions), with I² = 86.3%, indicating substantial heterogeneity
-
The authors should emphasize that this value represents a descriptive trend rather than a robust effect size.
-
A sensitivity analysis (e.g., excluding outliers or stratifying by laser type) could clarify the sources of heterogeneity.
-
-
Quality of included studies
-
Only one RCT was identified,
-
The remaining studies are small observational series, often with inconsistent outcome definitions and limited follow-up.
-
The discussion correctly acknowledges this but could benefit from a clearer summary table comparing risk-of-bias domains across studies.
Definition and pooling of outcomes
-
The term “significant response rate (>50% improvement)” merges complete and partial responses, which differ substantially in clinical meaning.
-
Adverse events are presented narratively; a pooled incidence (even if descriptive) could provide a more quantitative summary.
-
-
Author Response
The reviewers' notes:
Comment 1: " The quantitative synthesis includes only six studies (79 patients, 371 lesions), with I² = 86.3%, indicating substantial heterogeneity. The authors should emphasize that this value represents a descriptive trend rather than a robust effect size. A sensitivity analysis (e.g., excluding outliers or stratifying by laser type) could clarify the sources of heterogeneity."
Response 1: We thank the reviewer for this important observation. Therefore, we have now added a leave-one-out sensitivity analysis to assess the influence of individual studies on the pooled estimate. The methodological details are described in Section 2.6 (Data Analysis and Synthesis), and the results are presented in Section 3.6 (Meta-analysis). This analysis confirmed that no single study significantly altered the overall trend.
Comment 2: " Only one RCT was identified. The remaining studies are small observational series, often with inconsistent outcome definitions and limited follow-up. The discussion correctly acknowledges this but could benefit from a clearer summary table comparing risk-of-bias domains across studies."
Response 2: We appreciate the reviewer’s suggestion. However, we believe that the current Baseline characteristics and quality assessment table (Table 1) already presents detailed information for each included study. Adding an additional domain comparison summary table may introduce unnecessary redundancy and reduce the visual clarity of the manuscript.
Comment 3: " The term ‘significant response rate (>50% improvement)’ merges complete and partial responses, which differ substantially in clinical meaning. Adverse events are presented narratively; a pooled incidence (even if descriptive) could provide a more quantitative summary."
Response 3: We thank the reviewer for this insightful comment. Regarding adverse events, our original intention was to perform a quantitative analysis summarizing their incidence across studies. However, most of the included articles reported adverse events only descriptively, without sufficient numerical data to permit pooled analysis. Therefore, we opted to present a narrative synthesis of adverse events, as noted in the Results and Discussion sections.
Reviewer 2 Report
Comments and Suggestions for Authors
This is a highly relevant and well presented manuscript. One area of concern: lymphedema discussion. You cite 3 papers that mention that laser helped with lymphedema yet you did not discuss the extent or grade of lymphedema in these papers. It is dangerous to imply that laser is helpful in lymphedema in that extensive lymphedema is considered to indicate systemic disease requiring chemo. Either discuss grade of lyphedema, take out that section and /or make reference to this in 4th paragraph of discussion
Author Response
Reviewer 2-
Comment 1: “This is a highly relevant and well-presented manuscript. One area of concern: lymphedema discussion. You cite 3 papers that mention that laser helped with lymphedema yet you did not discuss the extent or grade of lymphedema in these papers. It is dangerous to imply that laser is helpful in lymphedema in that extensive lymphedema is considered to indicate systemic disease requiring chemo. Either discuss grade of lymphedema, take out that section and /or make reference to this in 4th paragraph of discussion.”
Response 1: We thank the reviewer for pointing this out. We agree that the absence of details regarding the grade or extent of lymphedema limits the interpretation of this observation. Accordingly, we revised the discussion (fourth paragraph) to clarify that the included studies did not specify the initial severity of lymphedema in these cases, and the statement was accordingly modified to avoid overinterpretation.